# Positioning Localities for Vague Spatial Location Description: A Supervaluation Semantics Approach

**Peng Ye** [1,2,3] (ID)**, Xueying Zhang** [1,4,]*****, Chunju Zhang** [5,6] **and Yulong Dang** [1,4]

1 Key Laboratory of Virtual Geographic Environment, Nanjing Normal University, Ministry of Education, Nanjing 210023, China; 007839@yzu.edu.cn (P.Y.); 211302159@njnu.edu.cn (Y.D.)
2 Urban Planning and Development Institute, Yangzhou University, Yangzhou 225127, China
3 College of Civil Science and Engineering, Yangzhou University, Yangzhou 225127, China
4 Jiangsu Center for Collaborative Innovation in Geographical Information Resource Development and Application, Nanjing 210023, China
5 School of Civil Engineering, Hefei University of Technology, Hefei 230009, China; zhangspring@hfut.edu.cn
6 Key Laboratory of Urban Land Resources Monitoring and Simulation, Shenzhen 518034, China
***** Correspondence: zhangxueying@njnu.edu.cn; Tel.: +86-025-8589-1561

**Abstract:** In the big data era, spatial positioning based on location description is the foundation to the intelligent transformation of location-based-services. To solve the problem of vagueness in location description in different contexts, this paper proposes a positioning method based on supervaluation semantics. Firstly, through combing the laws of human spatial cognition, the types of elements that people pay attention to in location description are clarified. On this basis, the source of vagueness in the location description and its embodiment in the expression form of each element are analyzed from multiple levels. Secondly, the positioning model is constructed from the following three aspects: spatial object, distance relation and direction relation. The contexts of multiple location description are super-valued, respectively, while the threshold of observations is obtained from the context semantics. Thus, the precisification of location description is realized for positioning. Thirdly, a question-answering system is designed to the collect contexts of location description, and a case study on the method is conducted. The case can verify the transformation of a set of users' viewpoints on spatial cognition into the real-world spatial scope, to realize the representation of vague location description in the geographic information system. The result shows that the method proposed in the paper breaks through the traditional vagueness modeling, which only focuses on spatial relationship, and enhances the interpretability of semantics of vague location description. Moreover, supervaluation semantics can obtain the precisification results of vague location description in different situations, and the positioning localities are more suitable to individual subjective cognition.

**Keywords:** location description; vagueness; supervaluation semantics; positioning locality

## 1. Introduction

Since ancient times, human beings have never stopped exploring their own living environment. The mental ability of human beings to learn and understand the environment is known as spatial-cognitive ability. Through spatial cognition, people can analyze and study the occurrence, cause and effect and tendency of things and phenomena. It is helpful to inspire thinking in images in scientific exploration and belongs to the engine of inspiring creative thinking [1]. Location is one of the basic and essential features in spatial feature dimensions [2]. The spatial location can answer the major question about "where" and provide the spatial reference for the answer to other questions in geography [3]. Location description is the natural language expression of human spatial cognition. It is used to illustrate the distribution difference between space entities, which makes people interact with the real environment [4]. Since natural language is the primary and basic means of information transmission in human society, location description is an important

medium for transmitting spatial location information in human communication [5,6]. At the present stage, artificial intelligence-induced changes are spreading to all walks of life. The mutual perception among environment, human and machine is also developing toward the direction of intelligence and convenience for people to use [7]. Natural language has become an important way to eliminate the interaction barrier between human and machine [8]. In the big data era, location based services (LBS) based on natural language interaction through speech or text have become a trend. This can promote the next generation of geo-information platforms to intelligent, ubiquitous development of the overall transformation.

Location description exists qualitatively or semi-quantitatively. Although it provides abundant clues for positioning, it also has vagueness generally [9]. This kind of vagueness includes not only the spatial object itself, but also the differences of individual spatial cognition and the description methods of natural language [10,11]. At present, the research on the vague spatial location description mainly adopts the two-valued logic modeling and fuzzy logic modeling. (1) Typical two-valued logic modeling includes the cone model, direction matrix model, Voronoi diagram-based model, detail direction relation expression model and so on [12–14]. According to the model structure, the space is rigidly divided into several regions. In the objective world, the transition of space in the adjacent direction is continuous and smooth. If the "black or white" model is adopted, the transition between the different result sets is jump and non-smooth [15,16]. Therefore, accurate description and processing of uncertainty often result in information loss. (2) Fuzzy logic uses membership functions (MF) to explain the degree to which each point in the space belongs to a fuzzy position [17,18]. As usual, MF is two-dimensional [19]. In order to determine the MF of a spatial location, it is necessary to use cognitive experiments, geographic information retrieval, remote sensing image classification and other methods [20–23]. Due to the fuzzy logic being linear order, every membership assignment can be compared with each other, whereas it is not suitable for multi-dimensional fuzzy problems.

Supervaluation semantics is a kind of non-binary logic used to explain "vagueness" in philosophy [24,25]. The core idea is to introduce indefinite values other than true and false for interpretation of vague expression which is difficult to judge authenticity [26]. For instance, the park is near the square, which is true in some cases and false in others. Thus, "near" is a boundary case that can be true or false. Supervaluation semantics can describe the relationship among uncertainty, certain true and certain false more precisely and intuitively. Besides, it also keeps the classical logic to some extent [27]. Compared with other solutions to the vagueness problem, supervaluation semantics is easier to be understood and accepted [28].

The quantitative direction, coordinate and region can be represented directly by points, lines and polygons in the geographic information system (GIS), whereas it is difficult for GIS to directly deal with the qualitative location description with vagueness. In fact, location descriptions are widely found in witness records, social media, historical documents and so on. Therefore, it is necessary to analyze the spatial information in vague location description so that it can be represented in GIS. This paper presents a positioning method based on supervaluation semantics. This method can transform the location description containing a set of spatial cognition views into real-world spatial scope and realize the representation of vague location description in GIS. The main innovation of this study is reflected in the following two aspects:

(1) Construction of vague location description representation model considering multi-factors. Based on the cognitive mechanism of spatial location, a unified framework of spatial location description is proposed, which defines the following three cognitive factors: reference frame, spatial object and spatial relationship. The framework is used to illustrate the vagueness of different cognitive and abstract levels in the location description. Different from the traditional spatial information modeling, which focuses on spatial relationship, this paper establishes the vagueness relation and influence among different information factors by the strategy of multi-factors representation. It not only combs the source of vagueness in location description

more comprehensively, but also enhances the interpretability of semantics of vague location description.

(2) A supervaluationist theory-based positioning method for vague spatial location description is proposed. Based on the basic principles of supervaluation semantics, precisification models of vague predicates are constructed for spatial object and spatial relationship. From the three aspects of extension, anti-extension and penumbra, the threshold range of the cut-off point in the precisification model is set according to the location description context, so as to obtain precisification results in different situations. Location description as the expression of the result of personal spatial cognition is strongly subjective. In this method, different contexts are supervalued separately, making the inferred spatial location in real world more suitable for personalized subjective cognition.

The following sections are expanded as follows: Section 2 summarizes the research status of related work; Section 3 analyzes the expression form of vague location description from the three aspects of reference frame, spatial object and spatial relationship; Section 4 constructs a positioning model oriented to vague location description based on supervaluation theory; Section 5 designs a question-answering system and conducts a case study; Section 6 presents the conclusions and future work.

## 2. Related Work

### 2.1. Vague Location Description and Positioning Method

The spatial location information, which is produced by spatial entities with different spatial distribution positions, is closely related to human's understanding and transformation of the objective world [29]. During thousands of years of evolution, human beings have created plenty of language symbols and formed a linguistic space system to describe various spatial features between objects, involving specific expression models and spatial elements [30]. In natural language, reference object, target object and spatial relation are regarded as basic elements to express spatial location information [31]. Then, it forms a relative and qualitative description by combining relevant words, phrases and sentence forms [32]. The description of spatial location in natural language has ambiguity and vagueness, which are similar but different [33]. For ambiguity, take the "convenience store on the street" as an example. There is more than one convenience store on the street, but the "easternmost one" can eliminate ambiguity by adding information. For vagueness, take "the restaurant near the hospital" as an example. The distance from some restaurants to the hospital is uncertain whether it is near or far, which satisfies the vague nature of natural language. In this paper, the vagueness of location description is mainly discussed.

Positioning localities for location description is to establish the mapping relationship between location information in natural language and spatial location in real world. Thus, it is inevitable to deal with vague location descriptions in positioning. At present, it mainly includes the method based on the two-valued logic modeling and the method based on fuzzy logic modeling. For the two-valued logic modeling, it is mainly oriented to the spatial relations such as distance, direction and topology. By artificially dividing the space into different regions, the transformation relationship between the description of the specific locality and the corresponding location of the real world is constructed [34]. The modeling of direction relation mainly includes the cone-shaped model, triangular model and minimum bounding rectangle model (MBR) [35,36]. The space is divided into 4 directions, 8 directions, 12 directions and so on. The modeling of distance relation is based on the method of "point-radius", which is divided the distance into different regions such as very far, far, near and very near. The modeling of topological relation mainly includes a 4-intersection model and 9-intersection model. Furthermore, a 9-intersection model based on dimension extension and a 9-intersection model based on the Voronoi graph are derived [37].

Fuzzy logic assumes that the degree to which a variable belongs to a fuzzy set can be expressed as a value between 0 and 1, hence the corresponding membership function can

be established. The function is expressed as z = f (x, y), where z represents the membership value at (x, y) [38]. Currently, the membership function is an effective model to describe the vagueness of the spatial scope of geographical entities. For instance, based on the division of the 4-direction model, a vague description model of vagueness direction relation is proposed [39]. Moreover, according to the principal direction relation, the vagueness of spatial direction relations in different cognitive scenarios is discussed, which provides a realistic basis for the vague description of direction relation [40]. Fuzzy logic can not only represent the irregular changes of fuzzy membership relations, but also facilitate the calculation of fuzzy sets in different spatial ranges [41]. However, the existing membership function focuses on the presentation of general spatial cognitive results, ignoring the factors that influence the location description. In the construction of the membership function, the selection of the interviewees, the reliability and other factors would affect the rationality of the membership value result. In addition, the knowledge base method [42], the geographic information retrieval method [43] and the remote sensing image classification method [44] are also used to obtain the membership value.

In addition to the above two main positioning methods, the point-radius method, the egg-folk model and the spatial clustering method are also proposed. (1) The point-radius method uses a point and a circle with a certain radius around the point to describe the location [45]. This method synthesizes all uncertainties into a radius, which is not only simple but also convenient for data storage. Nevertheless, because the coordinates of all objects are represented by points, the actual shape and size of the objects are not considered. (2) The egg-folk model is based on the minimum and maximum range to describe the uncertain region of the geographic entity. The folk represents the region that belongs exclusively to a geographical entity, while the egg represents the region that may belong to a geographical entity [46]. The egg-folk model is relatively simple and supports the interpretation of some important inferences involving the relationship between vague spatial ranges. However, the simplified method of geometric shape in the model is too rough to reflect the actual spatial distribution of objects. (3) The spatial clustering method divides a collection of abstract objects into multiple classes composed of similar objects. Numerous methods such as kernel density estimation (KDE) [47], density-based spatial clustering of applications with noise (DBSCAN) [48], and support vector machine (SVM) [49] have all been used to spatialize fuzzy geographic entities. For instance, point of interest (POI) data is applied to spatial range identification of urban fringe areas [50]. Different spatial clustering algorithms have different adaptability, which leads to the type of selection algorithm will affect the final positioning results. In general, many scholars have proposed methods for spatial positioning based on location descriptions. Various methods have advantages in different application fields, but there are also limitations that need to be further improved. The characteristics of the main positioning methods are shown in Table 1.

**Table 1.** The characteristics of the main positioning methods.

| | Type | Description | Advantage | Disadvantage |
|---|---|---|---|---|
| Two-valued logic method | Cone-shaped model | The reference object is replaced by the reference object's center of mass, and the plane space is divided into four directions of east, south, west, and north around the center of mass of the reference object. | The model is simple, and the regions are clearly divided. | Inability to handle entanglement, intersection, horseshoe shape, etc., sometimes leads to misjudgment. |

**Table 1.** *Cont.*

| Type | | Description | Advantage | Disadvantage |
|---|---|---|---|---|
| | Triangular model | The expansion of the four-direction cone-shaped model and the eight-direction cone-shaped model. | The triangular model takes into account the influence of the shape and size of the space target on the spatial relationship to a certain extent. | When the distance of the space target is far, the ability to distinguish the directional relationship is weak. |
| | Minimum bounding rectangle model | The projections of the two targets on the X and Y axes are used to establish the smallest rectangle to approximate the direction relation of the original target. It is an extended model of the triangular model. | Taking into account the shape and size of spatial entities, with the help of the reflexivity of the semantics of orientation relations. | The use of semantic reflexivity to accurately describe the positional relationship has limitations, and it cannot be described formally and judged effectively. |
| | 4-intersection model (4IM) | Based on the point set topology, it is defined by the intersection of the boundary and the inner point set, and divided according to its content. | Because it only uses the "empty" and "non-empty" of point set intersection to distinguish the relation, the method is simple. | There are many cases that are clearly distinguishable by people, but the model is powerless. |
| | 9-intersection model (9IM) | Introduce the "complement" of the point set and construct a 9-intersection spatial relationship model consisting of the boundary, the interior, and the complementary point set. | The 9-intersection model improves the 4-intersection model and enhances the uniqueness of the plane-line and line-line spatial relationships. | There is not much improvement in the representation of the spatial relationship of plane-plane, point-point, point-line, and point-plane. |
| | 9-intersection model based on dimension extension | Use dimension expansion method to expand 9IM. The dimensionality of the intersection between the boundary, interior and complement of the point, line, and plane is used as the framework for the description of the spatial relationship. | It is helpful to classify various spatial topological relations more accurately. | The operation complexity of the model is high. |
| | 9-intersection model based on Voronoi graph | The Voronoi region is used to replace the "complement" of the spatial target in the 9IM, and a nine tuples model of spatial relationships based on Voronoi is developed. | It can distinguish the neighboring and separated relationships of spatial objects, avoiding the difficulty of calculating the external relationships of spatial objects. | The operation complexity of the model is high. |
| Fuzzy logic method | Cognitive experiment | A group of people were selected to judge the vagueness of the objects studied in the form of questionnaires. | The combination of geography and psychology embodies the subjectivity of vagueness factor. | The cost of collecting experimental data is high, and the reliability of sample group affects the accuracy of results. |

**Table 1.** *Cont.*

| Type | | Description | Advantage | Disadvantage |
|---|---|---|---|---|
| | Knowledge base | The spatial relationship between the observer and the object is represented by spatial knowledge, and the mapping relationship between "phrases" and "regions" is established. | Summarize and analyze the experience of experts in the field and existing research results to form a spatial knowledge base. | The type and breadth of spatial knowledge involved in the knowledge base are limited. |
| | Geographic information retrieval | Through the analysis of the geographic content contained in the web page, the scope of a vague entity can be determined. | There is no need to conduct a large-scale questionnaire survey, and it is more convenient for the membership function of multiple geographic elements. | The selected search concept is prone to have the characteristics of population density or other tendencies, and the experimental results are prone to bias. |
| | Remote sensing image classification | Ground feature classification method based on remote sensing image. | The modeling process is clear. | It is only applicable to geographical elements that can be distinguished by remote sensing, and the accuracy of the vague range is directly restricted by the classification accuracy. |
| Other method | Point-radius method | A point and a circle with a certain radius around the point to describe the location. | The model is simple and convenient for data storage. | the coordinates of all objects are represented by points, the actual shape and size of the objects are not considered. |
| | Egg-folk model | Based on the minimum and maximum range to describe the uncertain region of the geographic entity. | Supports some explanations of important inferences involving the relationship between vague spatial scopes. | It is unable to deal with the complex constraints that a region may expand between its maximum and minimum values, and it is difficult to reflect the actual spatial distribution of objects. |
| | Spatial clustering method | Clustering fuzzy place-name data, and the range of convex hull of the cluster with the most points is the approximate space range of the fuzzy place names. | No assumptions are attached to the data distribution, and the characteristics of the distribution range of spatial data are studied from the data sample itself. | The choice of spatial clustering algorithm is critical to the vague range result. |

### 2.2. Supervaluation Theory and Its Application in Geography

In the book "the approach to science", published in 1958, Henry K Mehlberg, logical positivism, proposed the basic idea of supervaluation based on statements of uncertainty arising from the vagueness of ordinary language [51]. In 1966, Van Franssen formally proposed the theory of supervaluation to deal with the semantic problems caused by sentences containing singular terms without reference [52]. Supervaluation semantics is developed based on classical semantics. Because the theory of supervaluation holds that some sentences are "neither true nor false", the truth-value gaps are proposed [53]. Therefore, supervaluation is seen as a kind of three-valued logic. In supervaluation semantics, "true" refers to supertrue, "false" refers to superfalse, and otherwise belongs to the truth-value gaps [54]. Supervaluation semantics use precisification to explain vagueness, minimizing

and eliminating the penumbra of vague predicates, and gradually expanding its extension and anti-extension.

The theory of supervaluation is mostly applied in philosophy, and it was introduced into geography at the beginning of the 2000s. In geography, the supervaluation theory is firstly applied to explain the fuzzy boundary of geographical entities. There are many vague predicates of space in the field of geography, such as "forest" and "grassland", "near" and "far", and so on [55–57]. This kind of vague predicate has no definite space region, and the boundary between predicates is difficult to define. Supervaluation theory holds that vague predicates are finite expressions in a particular context [58]. Based on cognitive logic, a set of contextual threshold parameters are super-valued. Then, a set of rules with this unconstrained threshold parameter is given. Because the supervaluation theory does not depend on specific values, it can be combined with the qualitative method seamlessly [59,60]. It is very important to explain the vagueness of geographical entities, and also to provide a reference for the discussion of the vagueness of spatial location description in this paper.

## 3. Spatial Location Description and Its Vagueness

### 3.1. From Spatial Cognition to Location Description

Spatial cognition is the ability of the human to recognize the existence, change mode and relative localities of various things and phenomena in the surrounding environment. Spatial cognition is also a process of processing all kinds of spatial information, including the relative location, spatial distribution and dependence of environmental things and phenomena, as well as their evolution with time [61]. Spatial cognition goes through the following three stages: "realistic space", "cognitive space" (including a series of processes such as perception, representation, memory, thinking, etc.) and "descriptive space" (Figure 1).

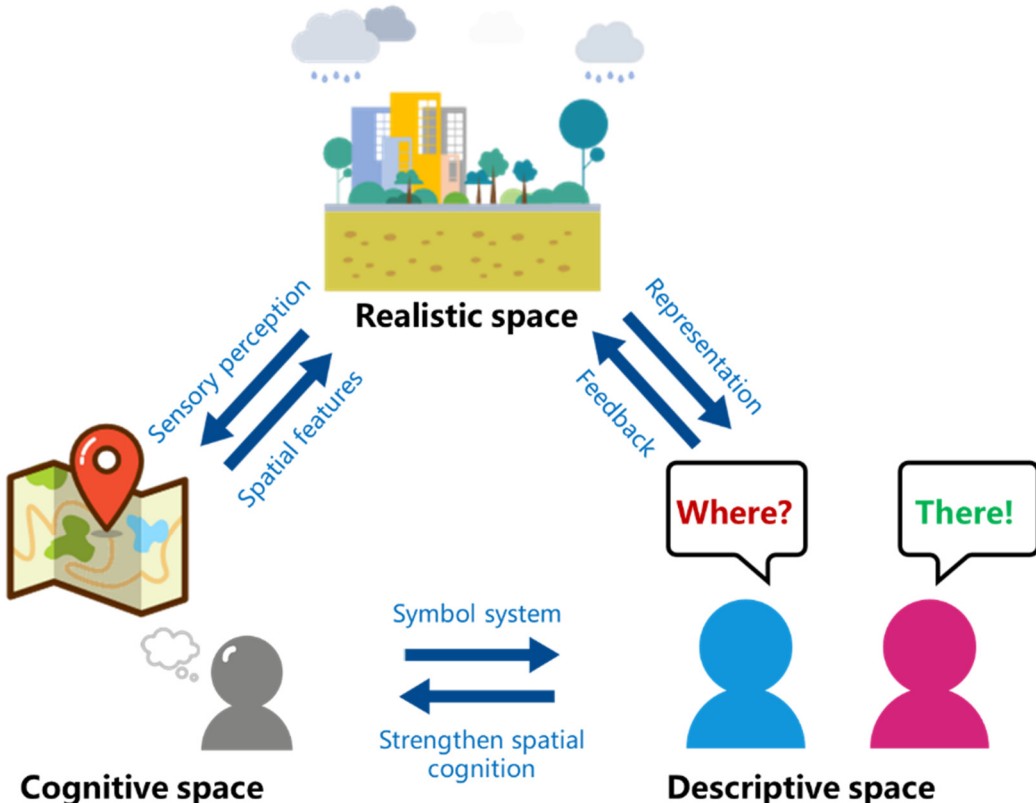

**Figure 1.** Different stages of spatial cognition.

Space is used to mark the difference of distribution among different things. The location is the characteristic of the place occupied by things in space. When one perceives the spatial location of a thing, it always refers to the relative location of the thing with another thing or other things. The thing will exist in the space where other things represent, and its relative location will change as it moves. Therefore, the location is the spatial scope occupied by the spatial object under the spatial reference frame, and manifests through the relative relations between spatial objects.

To communicate and transfer the spatial location acquired by cognition, natural language is used to describe the location. Each language has a set of words and syntactic structures that express various spatial locations. People can organize all kinds of languages to describe, narrate and explain the real world they perceive, and form corresponding phrases, sentence groups, paragraphs and chapters. From spatial cognition to location description, people perceive things and their relationships in the objective space based on individual cognitive ability and specific needs. Then, the language system is used to abstract cognitive results to elaborate. Therefore, the location description is the final presentation of the superposition of real space, spatial cognition and linguistic system.

### 3.2. Expression of Location Description

The location information described by natural language basically includes three forms. (1) Toponym. For example, "Yangzhou City", "Yangtze River" and other concrete names, as well as "red rooftop buildings" and other reference names. (2) Address. For example, "No.196, Huayang West Road, Hanjiang District, Yangzhou City", as well as telephone numbers, IP addresses and other generalized addresses. (3) Spatial assertions and their combinations. For example, "near Deji Plaza" or "east of Yangzhou University". Toponyms and addresses can be localized by means of geocoding. This paper mainly analyzes the expression characteristic of location description in spatial assertion form.

#### 3.2.1. Reference Frame

The reference frame is an important reference system for identifying different spatial objects and cognizing space. Reference frames can be classified into the following three types: absolute reference frame, relative reference frame and intrinsic reference frame. (1) The absolute reference frame is based on the fixed direction provided by gravity. (2) The relative reference frame divides the space by the front, back, left and right of the observer. (3) The intrinsic reference frame fixes the spatial reference to a background object other than the observer. In general, the reference frame is only the basis of location description based on spatial cognition, and people would not take the initiative to elaborate the reference frame in natural language. Although the reference frame is implicit in location description, different reference frames affect the selection of reference objects and the use of related terms (e.g., location words).

#### 3.2.2. Spatial Object

Spatial object, as the subject of spatial scope, is the abstract existence in which the objective things or phenomena occupy the location in the real world. Spatial objects exist in the frame of reference and are divided into reference objects and target objects. In natural languages, spatial objects are usually described in the following two ways: (1) The name of the spatial object. For instance, place name and its abbreviation, alias, point of interest (POI) name, landmark name, etc. (2) The attribute characteristics of spatial objects, including category, grade, size, matter, form and function, and so on. For instance, the six-story building with the red roof. When describing the location of target objects, multiple spatial objects may be selected as reference objects to enhance the precision of the description.

#### 3.2.3. Spatial Relationship

Spatial relationship constrains the spatial connection between the target object and other reference objects, including the following three basic types of binary spatial relation-

ships: topology, direction and distance. The distance relation has the strongest constraint on the spatial location, the direction relation is the second, and the topology relationship is the weakest. Thus, this paper focuses on the distance relation and direction relation. (1) The description of direction relation can be divided into absolute direction relation and relative direction relation. The description of relative direction relation (e.g., up, down, left, right, etc.) can be converted into an absolute direction relation (east, south, west, north, etc.) according to the observer's perspective. (2) The description of distance relation includes three ways. The first is the use of the "numeral + quantifier" form, such as "The distance between the house and the lake is about 500 m". The second is the use of specific adjectives, such as "far", "near", and so on. The third is to express the distance with the use of time, such as "walking 5 min".

### 3.3. Multi-Level Vagueness of Location Description

### 3.3.1. Vagueness of the Concept of Spatial Object

Spatial objects have spatial scopes with varying degrees of clarity. Clear spatial scope refers to the spatial scope with precise description and relatively accurate spatial location, such as national boundaries and other regime jurisdictions. The fuzzy spatial scope is the spatial scope with imprecise description and ambiguous spatial location, such as the transition zone from the existence to the non-existence and the transition zone between the adjacent regions. The former includes the range of typhoons, and the latter includes the boundary between forest and grassland. The vagueness of the concept of the spatial objects affects the vague expression of reference objects in the location description.

### 3.3.2. Vagueness in the Process of Spatial Cognition

Spatial cognition is usually not fully perceived from one observation point, and its perceived range is scale-dependent. The scale needs to adapt to the types of objects and needs of cognition, to choose and define a reasonable spatial scale for spatial cognition. For instance, when one needs to know the location of the "Lotus Pond Park in Yangzhou City", it is not accurate to only recognize the scale of the "central part of Yangzhou City", people would prefer to know that the park is located "opposite the east gate of Yangzhou University in Guangling District". Moreover, the scale of the reference object and target object need to match in the process of spatial cognition. For instance, in "Zhenjiang City is south of the Yangzhou University", there is a scale mismatch. Inapplicable and mismatched spatial scales will produce vagueness in the process of spatial cognition. In the location description, the appropriate scale is usually determined by the characteristics of the target object.

Due to the long-term living habits, cultural background, age and physiological and other aspects of the impact, spatial-cognitive ability is significantly different among individuals. Kevin Lynch, a leading urban planning theorist, points out that people's understanding of urban space mainly depends on their familiarity with the environment. For instance, local residents can accurately perceive the distance between the mall and the park to be about 500 m, while outsiders who are not familiar with the environment may wrongly estimate the distance between the two places to be 800 m. The abundant background knowledge is more advantageous to select the prominent reference object and the appropriate spatial relations for spatial cognition. On the contrary, the vagueness in the cognitive process may increase.

### 3.3.3. Vagueness Enhanced by Natural Language

The spatial location description in natural language belongs to qualitative expression, which will further enhance the vagueness of the result of abstract spatial cognition. On the one hand, there are vague predicates in natural languages, such as "near" and "north". These vague predicates lack clear boundaries when describing the spatial location. On the other hand, some adjectives and adverbs enhance the meaning of vague expressions, such

as "about", "almost", "close" and so on. Even if the precise expression of "50 m" is used quantitatively, the distance relation becomes vague after adding "about".

Therefore, although the sources of vagueness are various, the location description is the expression result of spatial cognition. Different levels of vagueness are superimposed on each other and are finally reflected in the location description (Figure 2). Understanding the vague features in location description is an important basis for positioning.

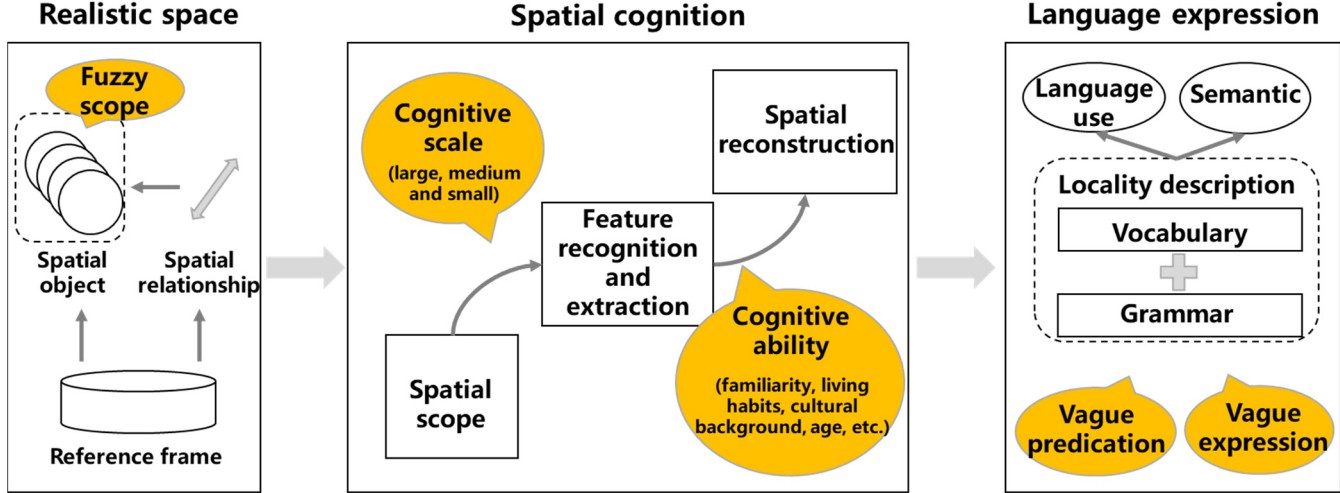

**Figure 2.** Source of vagueness in location description.

## 4. Positioning Model Based on Supervaluation Semantics

### 4.1. Basic Ideas of Supervaluationist Theory

Supervaluation is a typical semantic concept. Supervaluation semantics, also called supervaluationist theory, is developed based on classical semantics. The two-valued principle of classical logic only applies to clear statements, because classical semantics holds that true and false are exhaustive and mutually exclusive. According to supervaluationist theory, a sentence is true if and only if it is true in all ways of making it precise. This yields borderline case predictions that are neither true nor false, but classical logic is preserved almost entirely. The supervaluationist theory explains vagueness from the following points.

(1) Extension and anti-extension. Some lexical items, such as "forest", "far", "near" and so on, belong to the predicates, which are used in natural language to show the nature, characteristics or relationship between objects. Within a certain domain, the set of objects that can definitely be applied to a predicate is an extension of the predicate. On the contrary, the set of objects that cannot definitely be applied to this predicate is anti-extension. For instance, in the domain of "restaurants near the supermarket", the restaurant 500 m away is in the extension of the predicate "near", whereas the restaurant 2000 m away is in the anti-extension of the predicate "near".

(2) Penumbra. Beyond the extension and anti-extension is the boundary case of the predicate, which is the penumbra, also known as truth-value gaps. The existence of penumbra results in vague predicates that are neither true nor false. Because the supervaluation contains the truth-value gap, the classical two-valued principle is invalid for the supervaluationist theory. Following the previous instance, the restaurant 1000 m is the boundary case, thus the distance of 1000 m is in the penumbra of the predicate "near".

(3) Precisification. According to the supervaluationist theory, vagueness results from semantic incompleteness or indeterminacy of vague predicates, which makes the sentences containing vague predicates can be interpreted in different ways [62]. The use of more precise statements to explain each way is precisification. The aim of precisification is to minimize and eliminate the penumbra of vague predicates, and gradually enlarge the range of extension and anti-extension of predicates to make them

admissible precisifications, which is a reasonable and cognitive way of precisification. Under each admissible precisification, vague predicates are actually refined into precise predicates, and the sentences containing vague predicates can be assigned values in the way of classical logic. The purpose of supervaluation is to transcend the classical assignment that corresponds to each precisification.

(4)  Cut-off point. The cut-off point is a concept based on precisification, which is used to explain precisification in different contexts [63]. Following the previous instance, the 800m can be used as a cut-off point, with restaurants below 800m being included in the extension of "near", and all other restaurants in the anti-extension of "near". This is a kind of precisification of "near", and the result is a predicate without vagueness. By choosing different cut-off points, different precisification of the vague predicate can be obtained. It is important to note that the boundary case of the vague predicate is already determined. Therefore, any cut-off point of precisification should be appropriate and admissible, and not be inconsistent with non-boundary cases. Continuing the previous instance, it is not appropriate to use 400 m as a cut-off point for the precisification of the "near". Because this cut-off is within the known extension of the "near". Consequently, the cut-off point should be derived from the boundary case.

### 4.2. Positioning Model of Single Spatial Assertion

Spatial assertion expresses the location of the target object by describing the reference object and spatial relation. However, there may be vagueness in the natural language description of the reference object and spatial relation. Supervaluation is a classical theory to explain vagueness at the semantic level. The basis of the positioning model in this paper is supervaluation for multiple contexts of spatial assertion [64]. The aim of the positioning model is to establish the mapping relationship between vague location description and real spatial location. When there is only one set of the reference object, the target object and their spatial relationship, it is defined as a single spatial assertion.

### 4.2.1. Basic Framework of the Positioning Model

The single spatial assertion $V_o$ contains reference object description $V_r$, distance relation description $V_{dis}$ and direction relation description $V_{dir}$. $V_o$ is the combination of $V_r$, $V_{dis}$ and $V_{dir}$. The precisification of vague $V_o$ is dependent on context $c$. The model can be formalized as:

$$\{o | L(o) \wedge V(o, c)\}, V(o, c) \leftrightarrow n(c) \subseteq range(o) \tag{1}$$

$$n(c) = n(c_r) \cap n(c_{dis}) \cap n(c_{dir}) \tag{2}$$

In the formula, $L$ is the location of the target object $o$ in the real space. In the location description $V$ of $o$, the context $c$ with vague predicates is included. Context $c$ can be divided into the context $c_r$ of the reference object $r$, the context $c_{dis}$ of the distance relation $dis$, and the context $c_{dir}$ of the direction relation $dir$. The $range$ function represents the spatial range of the location of $o$. When the location $n(c)$ descripted in the context $c$ is within the range of $o$, $V(o, c)$ is true. To determine whether $V(o, c)$ is true requires that $V_r$, $V_{dis}$ and $V_{dir}$ should have precisification, respectively.

1.  Vague spatial object model

Vague predicates that express spatial objects in location descriptions are usually object names or antonomasia, which represent the types of spatial objects. Compared with the target object, the reference object should be a relatively familiar and representative spatial object among the users of the location description. Therefore, the user of location description can roughly grasp the existence scope of the spatial location of the reference object, but the description of the reference object is still vague, due to the difference in spatial cognitive ability. The precisification of vague $V_r$ depends on the context $c_r$. The model can be formalized as:

$$\{r | L(r) \wedge V(r, c_r)\}, V(r, c_r) \leftrightarrow n(c_r) \subseteq range(r) \tag{3}$$

In the formula, $L$ is the location of the reference object $r$ in the real space. In the location description $V$ of $r$, the context $c_r$ with vague predicates is included. The *range* function represents the spatial range of the location of $r$. When the location $n(c_r)$ described in the context $c_r$ is within the range of $r$, $V(r, c_r)$ is true. Taking the reference "A mall" as an example, the main rules for the precisification of $V(r, c_r)$ are defined as follows:

$$(\forall x)(\forall y)(\forall c_r)(\mathrm{mall}(x, y, c_r)) \leftrightarrow \Delta_{sld}(\mathrm{map}(x, y)) \subseteq \mathrm{in}(c_r)) \tag{4}$$

In the formula, $\mathrm{mall}(x, y, c_r)$ means the precisification of vague "mall" depends on the context $c_r$. The $\mathrm{map}(x, y)$ function maps the cognitive position $x$ in the location description to the real spatial location $y$, $\Delta_{sld}$ measures the distance between different location, and the *in* function maps the context to a value. The referent rule provides sufficient and necessary conditions for $mall(x, y, c_r)$. The necessary condition indicates that if the cognitive position $x$ matches the real location $y$ in a given context $c_r$ ($x$ is in the *range* of y), then in the $c_r$, the distance between $x$ and $y$ must be in a threshold value of in. The sufficient condition indicates that if the distance between $x$ and $y$ is a threshold value in a given context $c_r$, then $x$ matches $y$ in the $c_r$. However, the threshold for $\mathrm{in}(c_r)$ is not explicitly set, as it is also constrained by observations.

2.    Vague distance relation model

There are many vague predicates to express distance relation in location description, such as "very far", "near" and so on. The precisification of vague $V_{dis}$ depends on the context $c_{dis}$. The model can be formalized as:

$$\{dis|R(dis) \wedge V(dis, c_{dis})\}, V(dis, c_{dis}) \leftrightarrow n(c_{dis}) < val(x) \tag{5}$$

In the formula, $R$ is the spatial relation in the real space. In the location description $V$ of *dis*, the context $c_{dis}$ with vague predicates is included. The *val* function represents the distance between the reference object and the target object. When the distance $n(c_{dis})$ described in the context $c_{dis}$ is within *val*, $V(dis, c_{dis})$ is true. Taking the distance "near" as an example, the main rules for the precisification of $V(dis, c_{dis})$ are defined as follows:

$$(\forall r)(\forall o)(\forall c_{dis})(\mathrm{near}(r, o, c_{dis})) \leftrightarrow \Delta_{sld}(\mathrm{geo}(r), \mathrm{geo}(o)) < \mathrm{low}(c_{dis})) \tag{6}$$

In the formula, $\mathrm{near}(r, o, c_{dis})$ means the precisification of vague "near" depends on the context $c_{dis}$. The $geo(x)$ function maps a spatial object to its spatial location, the $\Delta_{sld}$ measures the distance between different location, and the low function maps the context to a value. The distance rule provides sufficient and necessary conditions for $near(r, o, c_{dis})$. The necessary condition indicates that if two objects are "near" to each other in a given context $c_{dis}$, then in the $c_{dis}$, the distance between the object-related geometry must be lower than a threshold value of *low*. The sufficient condition indicates that if the distance between two objects is lower than a threshold value of *low* in a given context $c_{dis}$, then the two objects are "near" to each other in the $c_{dis}$. $\mathrm{low}(c_{dis})$ is also constrained by observations.

3.    Vague direction relation model

There are also a lot of vague predicates in location description, such as "east", "left" and so on. The precisification of vague $V_{dir}$ depends on the context $c_{dir}$. The model can be formalized as:

$$\{dir|R(dir) \wedge V(dir, c_{dir})\}, V(dir, c_{dir}) \leftrightarrow n(c_{dir}) \subseteq exp(dir) \tag{7}$$

In the formula, $R$ is the spatial relationship in the real space. In the location description $V$ of *dir*, the context $c_{dir}$ with vague predicates is included. The *exp* function represents the direction between the reference object and the target object. When the direction $n(c_{dir})$

described in the context $c_{dir}$ is within the *exp*, $V(dir, c_{dir})$ is true. Taking the direction "west" as an example, the main rules for the precisification of $V(dir, c_{dir})$ are defined as follows:

$$(\forall r)(\forall o)(\forall c_{dir})(\text{west}(r, o, c_{dir})) \leftrightarrow \Delta_{deg}(\text{ang}(r, o)) \subseteq \text{in}(c_{dir})) \tag{8}$$

In the formula, west($r, o, c_{dir}$) means the precisification of vague "west" depends on the context $c_{dir}$. the ang($r, o$) function maps a spatial object to the rectangular coordinate system, the $\Delta_{sld}$ measures the azimuth of the target object to the reference object, and the *in* function maps the context to a value. The direction rule provides sufficient and necessary conditions for west($r, o, c_{dir}$). The necessary condition indicates that if the target object is located in the "west" of the reference object in a given context $c_{dir}$, then in the $c_{dir}$, the azimuth of the target object to the reference object must be in a threshold value of *in*. The sufficient condition indicates that if the azimuth angle of the target object to the reference object is in a threshold value of in in a given context $c_{dir}$, then the target object is in the "west" of reference object in the $c_{dir}$. in($c_{dir}$) is also constrained by observations.

### 4.2.2. Calculating Thresholds from Context-Dependent Observations

1. Threshold of vague spatial object

The threshold setting in the positioning model needs to be based on results of observations, which are dependent on the context. Set the context $c_1$ = "Yangzhou city of Jiangsu province", $c_2$ = "East gate of the mall where auntie went yesterday", and $c_3$ = "Zoo in the travel plan of the cousin who is in primary school", and observations vary. For different spatial objects, the observations are arranged as follows:

$$\text{city}(x_1, y_1, c_1) \wedge \text{mall}(x_2, y_2, c_2) \wedge \text{zoo}(x_3, y_3, c_3) \ldots \tag{9}$$

Based on the set of observations above, threshold settings may be in($c_1$) = (a, b), in($c_2$) = {($a_1, b_1$), ($a_2, b_2$), ($a_3, b_3$), ($a_1, b_1$)}, in($c_3$) = {($a_1, b_1$), ..., ($a_n, b_n$), ..., ($a_1, b_1$)}. The setting of these thresholds is affected by the object type, cognitive scale, cognitive ability, familiarity and modifiers (Figure 3). For instance, Yangzhou is a city, the area of jurisdiction is very broad. However, in the case of the distance between Yangzhou city and Shanghai city, the threshold of the spatial object Yangzhou is still set as the point coordinate because of its large scale. Comparatively speaking, the zoo threshold setting is very fine, belongs to the city territory relatively small-scale cognitive scene. Thus, the threshold value of the spatial object zoo can be set to the surface coordinates. When the relationship between the mall and the zoo is recognized, the threshold value can also be set as point coordinates because the scales of spatial objects are similar. In addition, individuals with high cognitive ability are more accurate in recognizing the location of familiar spatial objects. In content $c_2$, auntie has a strong spatial cognitive ability as an adult, and the time node of yesterday is also close to the present. Therefore, for the threshold setting of the east gate of the mall, a smaller range of buffers can be attached. The cousin, who is still in primary school, has relatively poor spatial cognition. Besides, his travel plan is arranged for places he has not been to. The thresholds of zoo are set more ambiguously, and the additional buffer zone is widened appropriately. Furthermore, for "near", "peripheral", "opposite" and other similar modifiers, will expand the threshold set in varying degrees of the buffer zone.

Setting thresholds for vague spatial objects requires consideration from the five dimensions in Figure 3. The specific threshold value can be determined in combination with practical application scenarios and individual cognitive habits. The threshold setting under different scenarios can refer to the expert experience and cognitive experiments. On the one hand, the reference range of cognitive ability is obtained by relevant scholars in spatial cognition research. On the other hand, the range of spatial objects is judged according to the results of cognitive experiments. The threshold setting of different individuals can grasp the real spatial cognitive state according to the question-and-answer method. In the question-and-answer method, questions can be set for the dimensions of vagueness, and the vagueness can be explained according to the quantifiable answer to set the threshold range.

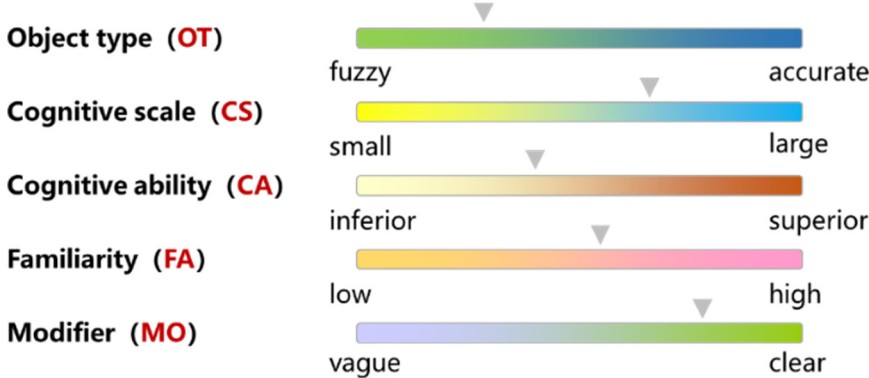

**Figure 3.** Threshold of observations of vague spatial object.

2.  Threshold of vague distance relation

In context $c_1$ = "10 min walk", $c_2$ = "5 min full run in the rain", and $c_3$ = "short-time biking", the differences of observations in distance relations are significant. For the reference object *r* and target object *o*, the observations are arranged as follows:

$$\text{near}(r_1, o_1, c_1) \wedge \text{near}(r_2, o_2, c_2) \wedge \text{near}(r_3, o_3, c_3) \dots \tag{10}$$

Based on the set of observations above, threshold settings may be $\text{low}(c_1)$ = 240 m, $\text{low}(c_2)$ = 750 m, and $\text{low}(c_3)$ = 800 m. These thresholds are set in a simple algorithm that can be trained in the literature [65]. The setting of these thresholds is affected by the movement mode, terrain environment, accessibility, weather condition and modifiers (Figure 4). For instance, in good weather, by car and another efficient means to travel to a place with flat terrain and road access, the location range of "near" is relatively broader. On the contrary, in bad weather, by walking and other inefficient means to travel to the place with rough terrain and road impassable, the location range of "near" is narrower.

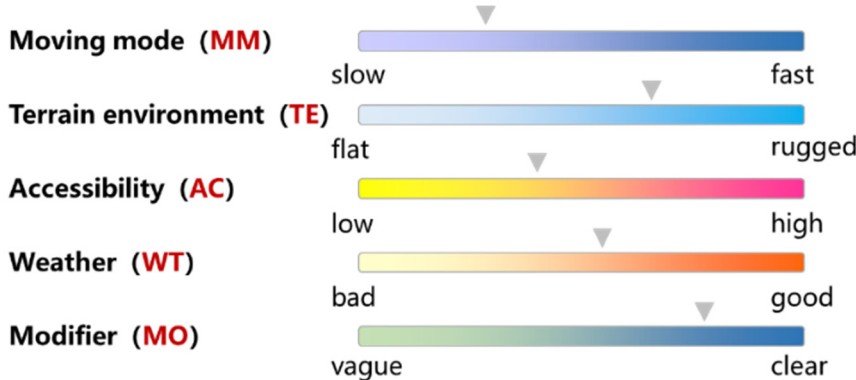

**Figure 4.** Threshold of observations of vague distance relation.

Setting thresholds for vague distance relation requires consideration from the five dimensions in Figure 4. The specific threshold value, which is affected by moving mode, terrain environment, accessibility and weather, can be determined by experimental test or expert experience as follows: on the one hand, in a real environment, or setting up a simulated environment that meets specific conditions, relying on monitoring equipment to detect the distance range in different environments; on the other hand, drawing on the

reference value of the spatial range obtained by relevant scholars. The specific threshold value, which is affected by the modifier, can be determined by the question-and-answer method as follows: explain the vagueness of modifier words based on quantifiable answers, establish the mapping relationship model between modifier words and vague, and set the threshold range on this basis.

3.   Threshold of vague direction relation

In context $c_1$ = "east", $c_2$ = "northeast", and $c_3$ = "ten o'clock direction", the differences of observations in direction relations are significant. For the reference object $r$ and target object $o$, the observations are arranged as follows:

$$\text{east}(r_1, o_1, c_1) \wedge \text{northeast}(r_2, o_2, c_2) \wedge \text{tenclock}(r_3, o_3, c_3) \dots \qquad (11)$$

Based on the set of observations above, threshold settings may be in($c_1$) = [−45°, 45°), in($c_2$) = [22.5°, 67.5°), in($c_3$) = [300°, 330°). The setting of these thresholds is affected by the cognitive dimensions, distance and modifiers (Figure 5). For instance, "east", "northeast", "ten o'clock direction" reflects the amount of space for division. When the distance between the target object and the reference object is close, the judgment of azimuth angle is more accurate. Vague modifiers such as "approximately" and "possibly" appear in the context, further expanding the scope of threshold setting.

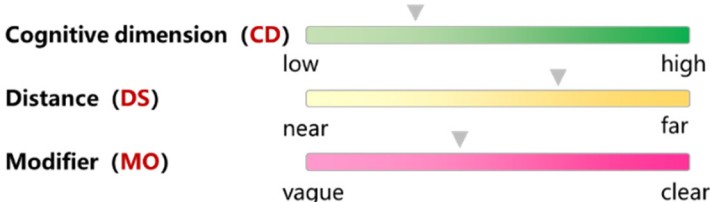

**Figure 5.** Threshold of observations of vague direction relation.

Setting thresholds for vague direction relation requires consideration from the three dimensions in Figure 5. The specific threshold value, which is affected by cognitive dimensions and modifier, can be determined by the question-and-answer method. The specific value, which is affected by distance, can be determined by experimental test or expert experience. The specific operation process can refer to the method of the threshold of vague distance relation.

4.2.3. Threshold Comparison between Different Contexts

Positioning models for spatial objects, distance relations and orientation relations all contain contextual parameters, such as mall($x$, $y$, $c$), near($r$, $o$, $c$) and east($r$, $o$, $c$). Define a predicate *ist*($p$, $c$) (or ¬*ist*($p$, $c$), means "true or false") to record a proposition $p \in P$ in a particular context $c$. Based on this, the predicates involved in the above positioning models can be transformed into *ist*(*mall*, $c_1$), *ist*(*running*, $c_2$), *ist*(*bike*, $c_3$), ¬*ist*(*raining*, $c_4$), ¬*ist*(*east*, $c_5$), etc. It should be noted that if the context $c_i$ does not contain a description of "raining", then neither the *ist*(*raining*, $c_i$) nor the ¬*ist*(*raining*, $c_i$) will be asserted. For predicates of the same type (as in the case of movement mode in a distance relation), relations between different contexts can be calculated after the observations of the predicates in the dependent context are obtained.

(1)   Equality ($c_i \approx c_j$). If the two contexts $c_i$ and $c_j$ are the same in terms of the observations, then they are likely to be equal and are recorded as $c_i \approx c_j$. In the instance above, $c_2! \approx c_3$.

(2)   General ($c_i \sqsupset c_j$). After ignoring observations, if there are:

$$(\forall p)\big(p \in \mathrm{P} \wedge \text{ist}(p, c_j) \rightarrow \text{ist}(p, c_i)\big) \cap (\forall p)\big(p \in \mathrm{P} \wedge \neg\text{ist}(p, c_j) \rightarrow \neg\text{ist}(p, c_i)\big) \quad (12)$$

The formula indicates that the contexts $c_i$ is at least as generic as the contexts $c_j$ (recorded as $c_i \sqsupseteq c_j$). If $c_i \sqsupseteq c_j$ but does not have $c_j \sqsupseteq c_i$, then $c_i$ is more general than $c_j$ (recorded as $c_i \sqsupset c_j$). In the instance above, $c_2 \sqsupset c_3$.

(3) Identifiability ($c_i \prec c_j$). Taking the predicate "near" as an example, if all is "near" in $c_i$ and is also "near" in $c_j$, then $c_i$ has at least the same identifiability as $c_j$ (recorded as $c_i \preceq c_j$). If $c_i \preceq c_j$ but does not have $c_j \preceq c_i$, then $c_j$ has a stronger identifiability (recorded as $c_i \prec c_j$). In the instance above, $c_2 \prec c_3$.

*4.3. Positioning Model of Compound Spatial Assertion*

Multiple single spatial assertions are combined to form compound spatial assertion. Single spatial assertion is the basic composition of compound spatial assertion. By expressing the relative location of the target object to several reference objects, the distribution scope of the target location can be described in more detail.

The compound spatial assertion $V_o$ of the target object contains multiple spatial relations between the different reference object and the target object, and the spatial assertion of each reference object involves the context $c_i$. Every $c_i$ contains reference object description $V_{r,i}$, distance relation description $V_{dis,i}$ and direction relation description $V_{dir,i}$. The precisifications of vague $V_{r,i}$, $V_{dis,i}$ and $V_{dir,i}$ are all depend on context $c_i$. The model can be formalized as:

$$\{o|L(o) \wedge V(o,c)\}, V(o,c) \leftrightarrow n(c) \subseteq range(o) \tag{13}$$

$$n(c) = n(c_1) \cap n(c_2) \cap \ldots \cap n(c_i) \tag{14}$$

$$n(c_i) = n(c_{r,i}) \cap n(c_{dis,i}) \cap n(c_{dir,i}) \tag{15}$$

In the formula, the definition of correlation parameter and function is the same as the positioning model of single spatial assertion.

For the precisification of the compound spatial assertion, firstly, the compound spatial assertion is divided into several single spatial assertions. Secondly, the precisification of single spatial assertion is used to positioning locality. Thirdly, the positioning results of each single spatial assertion are superimposed, and the intersection of them is the final positioning locality.

## 5. Case Study

In practical application, the positioning method oriented to vague location description is mainly used to remove the barriers between human and machine, and to serve various kinds of location services based on natural language interactive. Moreover, the positioning model proposed in this paper relies on abundant context. Therefore, a human–computer interactive Q&A (question and answer) system is designed to collect contextual data of location description in natural language. The Q&A system includes the following two basic modules: back-stage management and interactive system. In the back-stage management module, the administrator constructs and manages the questioning content and characteristic dimension (Figure 6). In the interactive system module, according to the settings in the background management module, the system asks pertinent questions about the characteristics of a specific spatial location, and then the user enters his (or her) understanding and cognition of the question in the about dialog box. Through the operation of the Q&A system, the contexts of location description are obtained. Furthermore, the positioning model for vague location description proposed in this paper is used, then the positioning localities are displayed by visualization methods.

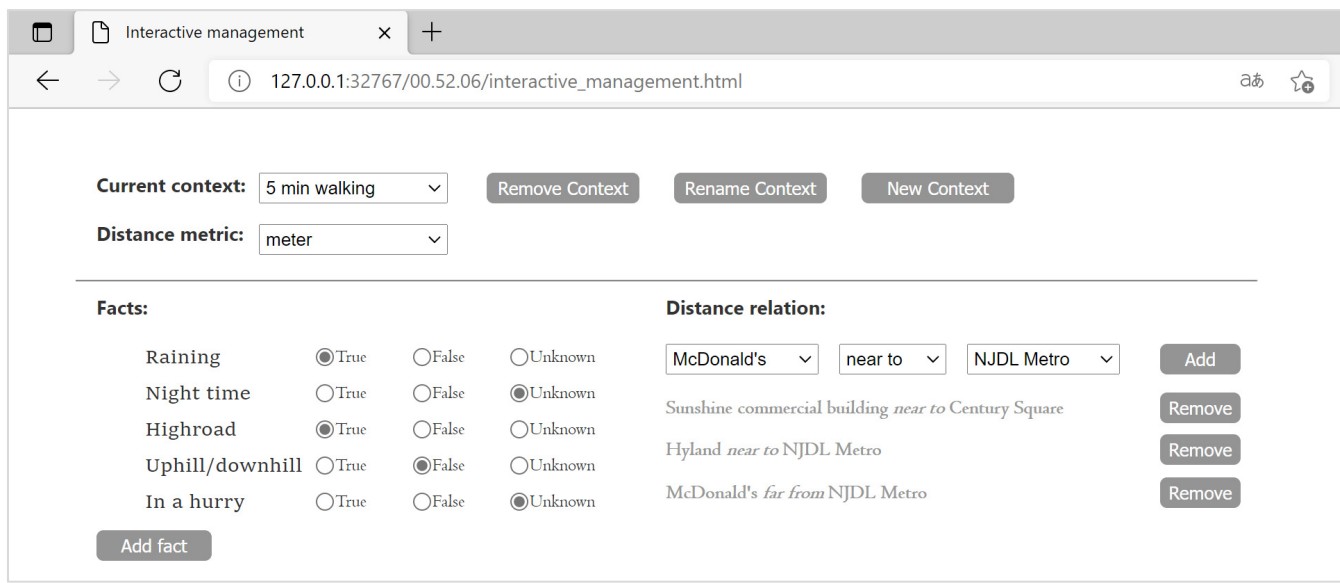

**Figure 6.** Interface diagram of back-stage management module in Q&A system.

In order to analyze the application effect of the method proposed in this paper, the Nanjing Road Walkway in Shanghai City in 2021 is selected as a case to illustrate the experiment. Nanjing Road is the most prosperous district in Shanghai, with a reputation for being known as "the first commercial street in China". A great deal of shops are interlaced, and it is typical to analyze the location description of this area. The plane graph of Nanjing Road is shown in Figure 7, and the location of some landmarks is noted.

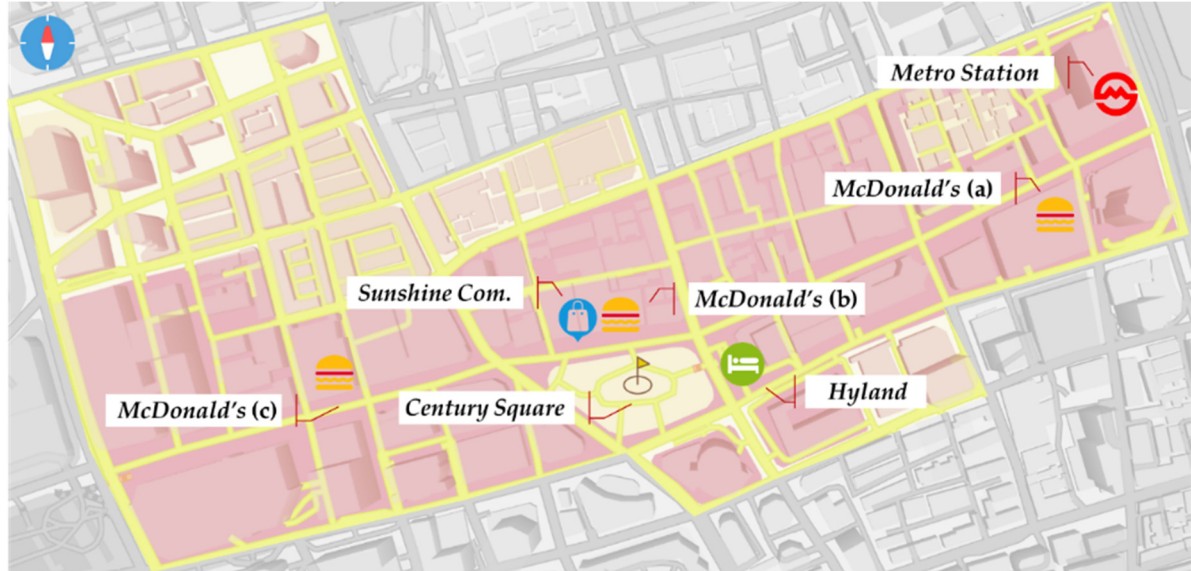

**Figure 7.** Nanjing Road Walkway and some landmarks.

*5.1. Positioning Localities Based on Single Spatial Assertion*

The location description text of the single spatial assertion is collected by the Q&A system. In this case, the user wanted to know which McDonald's on Nanjing Road are near to the metro station. The Q&A system asks the user about the moving mode, weather condition, accessibility and so on (Figure 8). Then, the context of the answer contains the user's cognition of the vague predicate "near" in different dimensions. The context of spatial assertion is super-valued, respectively, to select the McDonald's storefront which matches the location description.

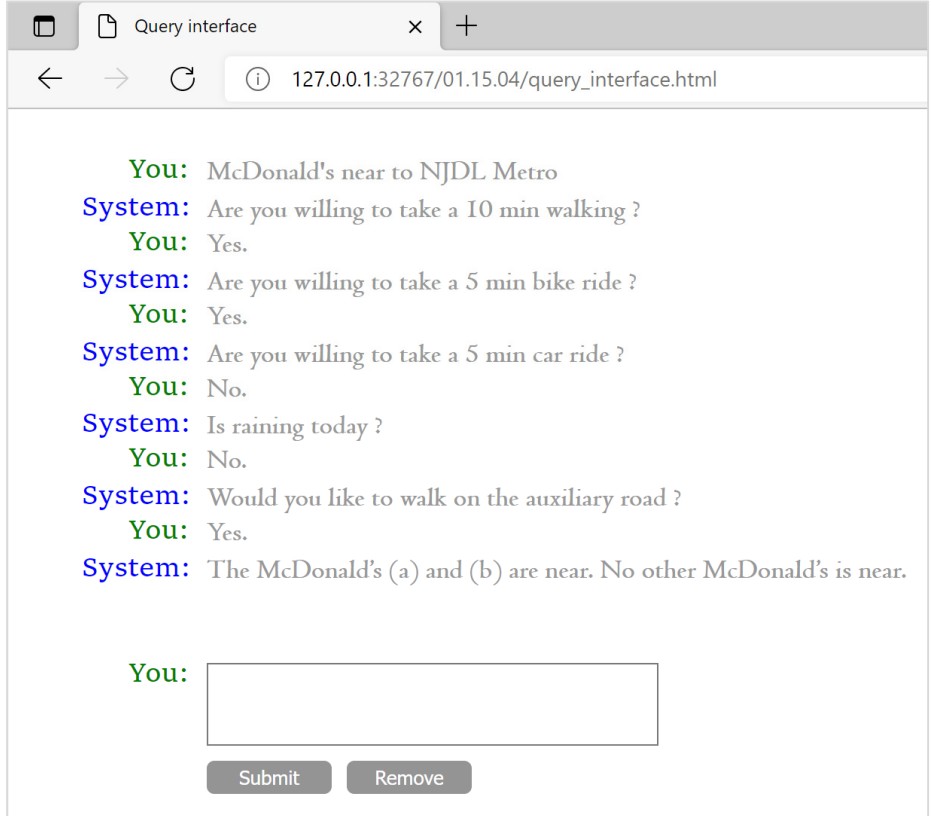

**Figure 8.** The contexts of single spatial assertion in Q&A system.

In the context of this single spatial assertion, the reference object is the East Nanjing Road Station metro station (NJDL Metro), the target object is the McDonald's storefront, the spatial relation involves the distance relation, and the direction relation is neglected. Therefore, the basis of positioning lies in the precisification of the distance relation between the metro station and the storefront. In terms of mobility, users prefer to walk or bike. When walking, the user's perception of "near" is no more than 10 min; when biking, the user's perception of "near" is no more than 5 min. However, the efficiency of moving mode is also affected by weather conditions. In terms of accessibility, there is a road link between the metro station and every storefront. According to Section 4.2.2, the context of Q&A system user's spatial cognition viewpoint is super-valued, and the trade-off points of distance relation are set according to the observation threshold (Table 2).

**Table 2.** Semantic analysis based on supervaluation of single spatial assertion.

| Factor | Asserted? | Context Semantics | Observations | Threshold Range |
|---|---|---|---|---|
| Movement mode | Yes | Walk, ≤10min | 300 m | ①$300 \times 1.0 \times 1.0 = 300$, the range is [0, 300 m] |
| | | Bike, ≤5min | 780 m | ②$300 \times 0.8 \times 1.0 = 240$, the range is [0, 240 m] |
| Terrain environment | No | - | - | ③$780 \times 1.0 \times 1.0 = 780$, the range is [0, 780 m] |
| Accessibility | Yes | High road | ×1.0 | ④$780 \times 0.8 \times 1.0 = 624$, the range is [0, 624 m] |
| | | Auxiliary road | ×0.8 | |
| Weather condition | Yes | Sunny | ×1.0 | |
| Modifiers | No | - | - | |

The actual location distribution of metro station and McDonald's storefront was compared with the threshold of observations. Storefronts (a), (b) and (c) are connected to the metro station. Storefronts (a) and (b) are 211 m and 668 m away from the metro

station, respectively, and storefront (c) is 913 m away from the metro station. The distance of storefront (a) accords with threshold ①, the distance of storefront (b) accords with threshold ③, whereas the distance of storefront (c) does not accord with every threshold. Furthermore, the walkway has restrictions on the movement mode, "metro station-storefront (b)" can only choose to walk, it does not meet the threshold of ①–②. Therefore, the distance between the metro station and storefront (a) is near (Figure 9).

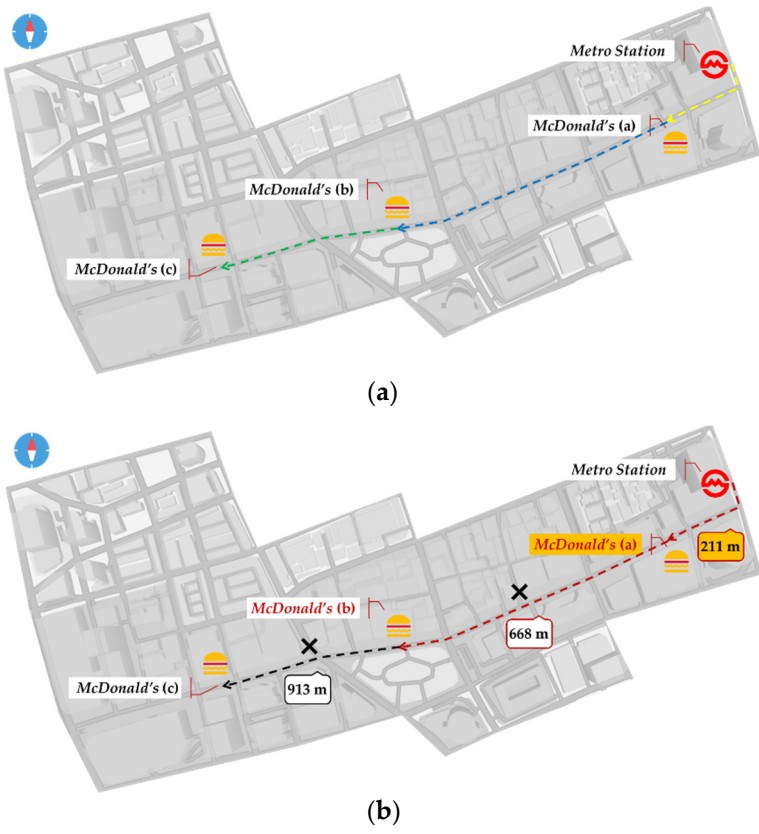

(**a**)

(**b**)

**Figure 9.** Positioning results of single spatial assertion. (**a**) is the schematic diagram of the path between every McDonald's storefront and the metro station. (**b**) is a schematic diagram of screening paths that meet the threshold according to the distance relation.

## 5.2. Positioning Localities Based on Compound Spatial Assertion

The location description text of the compound spatial assertion is collected by the Q&A system. In this case, the user wanted to know the location of the Century Square in the pedestrian block of Nanjing Road. The Q&A system asks questions from different reference objects and their spatial relations (Figure 10). Then, the context of answer contains cognition of the vague predicate "Century Square" in different dimensions. Several contexts of spatial assertion are super-valued, respectively to determine the spatial location of Century Square.

In the context of this compound space assertion, the reference objects are Hubei Road intersection, Sunshine commercial building and Hyland Hotel, the target object is Century Square, the spatial relationship involves distance relation and direction relation. Therefore, the basis of positioning lies in the precisification of the distance relation between Century Square and different reference objects. (1) Century Square is close to the east of the intersection of Hubei Road. Two vague predicates, "east" and "near", are involved in spatial assertion. Among them, "east" is used to indicate that the number of users dividing space azimuth is 4; because the target object is close to the reference object, azimuth judgment is accurate. The precisification of "near" can be referred to in Section 5.1, which also considers various factors such as moving mode, weather condition, accessibility and so on. (2) Century Square is opposite the Sunshine Commercial Building. The user's perception of the "opposite" is across the road, in the visible range. When setting the observation

threshold, Sunshine Mall is located near the road, Century Square is located on the other side of the road, and within 50 m of the roadside. (3) Century Square does not pass by Hyland Hotel. Due to moving along Nanjing Road, the course is from west to east. "Not pass" shows that the Century Square is located west of the Hyland Hotel and still divides the space into 4 directions. According to Section 4.2.2, the context of Q&A system user's spatial cognition view-point is super-valued, and the trade-off points of distance relation and direction relation are set according to the observation threshold (Tables 3–5).

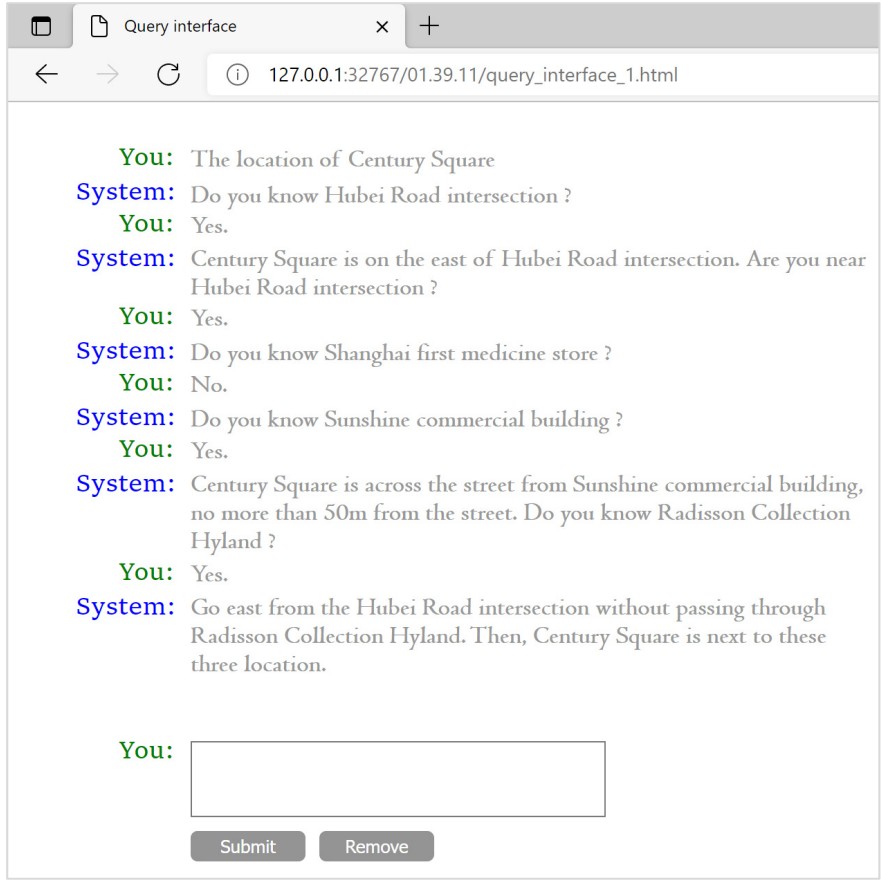

**Figure 10.** The contexts of compound spatial assertion in Q&A system.

**Table 3.** Semantic analysis based on supervaluation of spatial assertion of Hubei Road intersection.

| Type | Factor | Asserted? | Context Semantics | Observations | Threshold Range |
|---|---|---|---|---|---|
| Distance relation | Movement mode | Yes | Walk, ≤10min | 300 m | [0, 300 m] |
| | Terrain environment | No | - | - | |
| | Accessibility | Yes | Road | ×1.0 | |
| | Weather condition | No | Sunny | ×1.0 | |
| | Modifiers | No | - | - | |
| Direction relation | Cognitive dimensions | Yes | East, 4 dimensions | $[-45°, 45°)$ | $[-45°, 45°)$ |
| | Distance | Yes | Near | ×1.0 | |
| | Modifiers | No | - | - | |

**Table 4.** Semantic analysis based on supervaluation of spatial assertion of Sunshine Commercial Building.

| Type | Factor | Asserted? | Context Semantics | Observations | Threshold Range |
|---|---|---|---|---|---|
| Distance relation | Movement mode | Yes | Stand | [0, 50 m] | [0, 50 m] |
| | Terrain environment | No | - | - | |
| | Accessibility | Yes | Road | ×1.0 | |
| | Weather condition | No | - | - | |
| | Modifiers | No | - | - | |
| Direction relation | Cognitive dimensions | Yes | One side, 2 dimensions | 1 [1] | 1 [1] |
| | Distance | Yes | Near | ×1.0 | |
| | Modifiers | Yes | Opposite | - | |

[1] 0 is one side, 1 is the other side.

**Table 5.** Semantic analysis based on supervaluation of spatial assertion of Hyland Hotel.

| Type | Factor | Asserted? | Context semantics | Observations | Threshold Range |
|---|---|---|---|---|---|
| Direction relation | Cognitive dimensions | Yes | West, 4 dimensions | [135°, 225°) | [135°, 225°) |
| | Distance | No | - | - | |
| | Modifiers | No | - | - | |

According to the actual spatial distribution of the reference object and the threshold of the observations, the candidate location regions were determined (Figure 11). The angle of [45°, –45°] at the intersection of Hubei Road is "east" and is set as candidate location (a). Road junction with Hubei Road, and walk not more than 5 min distance is "near", both sides of the road area set as candidate location (b). There are three roads adjacent to the Sunshine commercial building, with the other side of the three roads area set as candidate location (c). The angle of [135°, 225°] at the Helen Hotel is "west" and is set as candidate location (d).

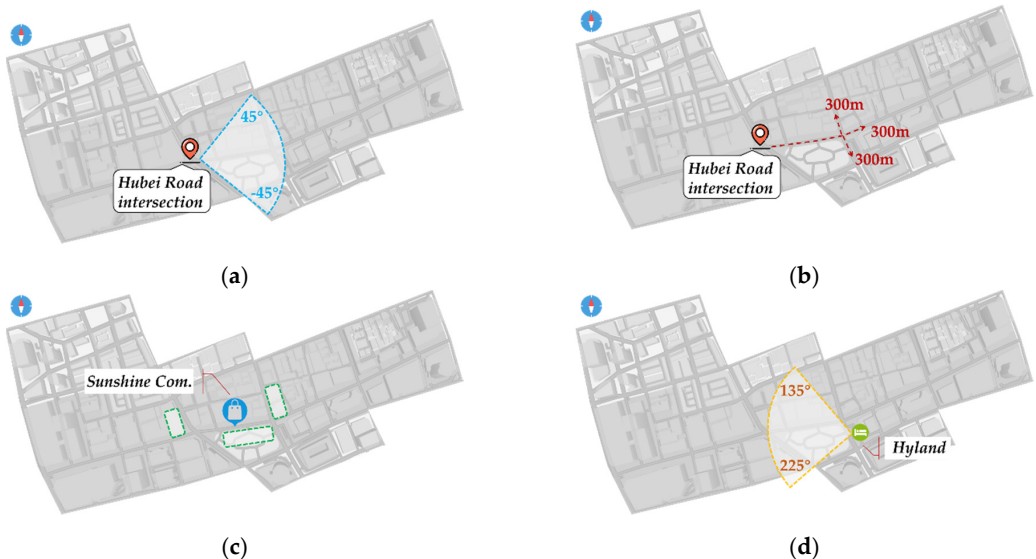

(**a**)  (**b**)

(**c**)  (**d**)

**Figure 11.** Positioning results of each compound spatial assertion. (**a**) is the schematic diagram of the direction relation with Hubei road intersection as the reference. (**b**) is the schematic diagram of the distance relation with Hubei road intersection as the reference. (**c**) is the schematic diagram of the distance and direction relations with Sunshine commercial building as the reference. (**d**) is the schematic diagram of the direction relation with Hyland hotel as the reference.

The candidate location (a)–(d), deduced from several spatial assertions, are superimposed to obtain the overlap region *D* (Figure 12). Then, *D* is the location of Century Square as described in the current spatial assertion. Compared with the real location *G* of Century Square, *D* is basically contained in *G*, and on the whole *D* is located in the northwest part of *G*. This is because the spatial relation is expounded from the west, the north and the east in choosing the reference object, and the spatial relation in the south is missing, which makes the result of super-valued in the south not precisification. In general, the spatial location of objects can be accurately obtained by supervaluation of multiple contexts of spatial assertion.

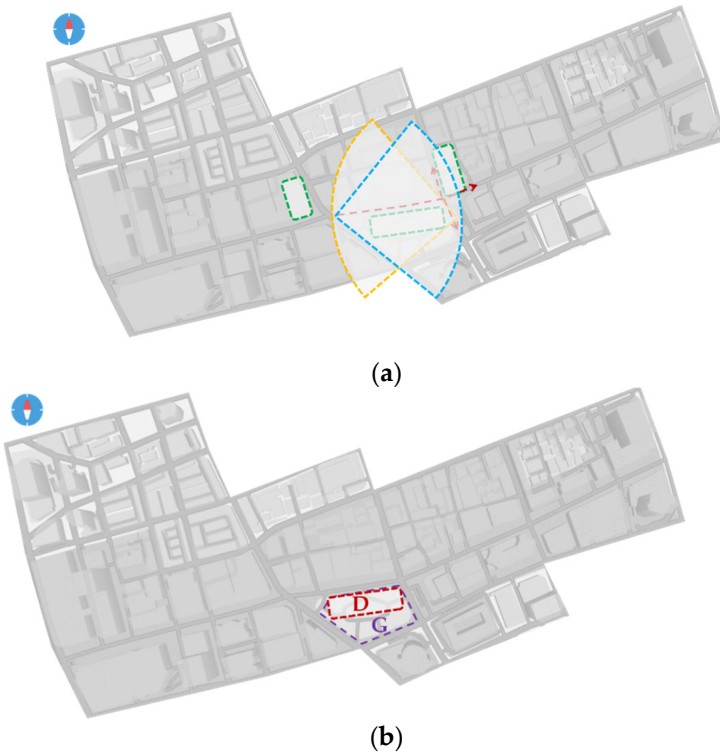

(**a**)

(**b**)

**Figure 12.** Positioning results of all compound spatial assertions. (**a**) shows all candidate location in Figure 11. (**b**) is the overlap of all candidate location. Besides, *D* is the location of Century Square determined by the candidate location, and *G* is the real location of Century Square.

*5.3. Discussion*

Natural language is the most basic means for human beings to transmit spatial location information, and the qualitative expression makes vagueness ubiquitous. In the current big data era, the design concept of intelligent LBS should embody the people-oriented purpose of information services and establish a harmonious interaction. The processing of location information should conform to the human perception and cognitive process. With the basic concept of spatial position, this paper clarifies the hierarchical relationship between spatial cognition and location description and the source of vagueness. The construction of spatial location representation model is an important precondition for the research of vague location description. On this basis, the supervaluationist theory is introduced into the field of spatial cognition. Based on the supervaluation semantics, the vague location description is transformed into precise spatial positioning. The advantages of supervaluation semantics are as follows: (1) Tautology is retained. The tautology of classical logic is still valid, that is, the two-valued logic of true or false remains unchanged. (2) The precisification results in a particular scene are unique. For instance, parks and playgrounds are located "far" from the boundary case. The playground is 100 m farther than the park. In the explanation of three-valued logic, the value of "park is far" and "playground is far" are the indefinite value *I* between true and false, and the conjunctive value is still *I*. Nevertheless, under the

supervaluation semantics explanation, because it is impossible to give the precisification of "playground is near" and "park is far". Then, the conjunctive form is false, the result is more in accord with the intuitional cognition. (3) For the multi-dimensional vagueness problem, since the precision structure is a partial order structure, it can express the incomparability between vagueness of different dimensions. In the case of "near", it can be short distance in a straight line, short distance in a journey, or short journey time. The vague predicate "near" is also a multi-dimensional feature of "near", and the "near" of different dimensions cannot be compared.

At present, the method based on two-valued logic modeling is still one of the main methods to analyze the location description semantics for spatial positioning—Umberto Straccia modeling for spatial relationships [66]. In this model, the traditional fuzzy description logics is extended to support fuzzy spatial reasoning. The existing research of two-valued logic modeling mainly focuses on the spatial relationship, and the spatial region is rigidly divided into several directions. However, the vagueness of the spatial direction relation means that the target object belongs to a certain direction concept with a certain degree of membership, which is a kind of multi-valued logic. Using precise methods to describe vagueness often results in information loss. Thus, the concept of fuzzy logic has begun to be introduced into the modeling of vague spatial direction. The membership function is the key to fuzzy logic modeling. Many scholars have proposed a variety of membership function construction methods. Edoh-Alove et al. obtained the user's tolerance for misuse of the spatial data set as the membership [67]. Dilo et al. used the mass distribution of fuzzy objects as the membership [68]. Cheng et al. discussed the influence of different spatial scales on the membership [69]. Xu et al. constructed subjective spatial knowledge as a knowledge base from the observer's perspective and provided knowledge support for determining the value of membership [70]. It can be seen that many studies attempt to investigate for the public, multi-scale, large-scale, in order to obtain the universal spatial cognitive law, and to provide a realistic basis for establishing the mainstream mapping relationship between vague description and real space.

In fact, everyone's perception of the spatial location is subjective, and the expression of location description is artificial. The same location description refers to different real spaces due to different scenes and individuals. The method proposed in this paper can describe the vagueness in the location description precisely and intuitively. On the one hand, it breaks through the traditional vagueness modeling that only focuses on spatial relationships. A more comprehensive consideration is given to the reasons and representations of the vagueness of the following three factors: reference frame, spatial objects and spatial relationships. On the other hand, by being super-valued to different contexts, the precisification results in different situations are obtained. Therefore, the positioning is adaptive to the subjectivity and artificiality of location description. In this method, the introduction of the precisification plays an important role. Among them, the choice of the cut-off point is the most critical. Location description is the result of the expression of human spatial cognition. While spatial positioning is a reverse process, which needs to understand the real spatial cognition through the location description. The cut-off point for the vague description actually needs to excavate the relevant understanding in the spatial cognition. With the help of appropriate cognitive experiments, vague semantics and psychology should be combined to reflect the subjectivity of cognitive activities. The supervaluation semantics matches the location description with each kind of actual scene. Through cognitive experiments, the threshold range of the cut-off points in different scenes is obtained, which becomes the final destination of the transformation from qualitative location description to quantitative spatial location.

The results of case study show that the existing models and methods can deal with the relatively simple description of location, whereas practical applications also need to extend the method to more complex scenarios. The complexity of location description increases its vagueness, and the factors considered in the process of precisification based on supervaluation semantics are more diversified. Therefore, the rules of the confirmation

of the conditional cut-off points and the multi-conditional compound computation are more complex, which need the specific prior and domain knowledge to guide the precisification. At present, knowledge graphs lead knowledge engineering of big data into a brand-new stage, which has advantages in computational and inferential aspects, and become an important tool of knowledge management and service [71]. Knowledge graphs transform the semantic parsing and precise positioning of the location description into intelligence, which is also facing challenges in the field of human–computer cooperation and complex reasoning.

## 6. Conclusions

Understanding location semantics from location description is the development direction of next–generation GIS location services. In this paper, a positioning method of vague location description based on supervaluation semantics is proposed. On the one hand, the vagueness of location description is analyzed from a multi-factor perspective. The traditional vague modeling of spatial relationship is extended to the modeling that takes into account the three aspects of reference frame, spatial objects and spatial relationships, which enhances the semantic analysis ability for the vagueness of location description. On the other hand, based on the spatial positioning of vague location descriptions, the spatial information in the description is converted into the spatial range that can be represented in GIS. Based on the supervaluation theory, the precisification results of location description in different situations are obtained, respectively, which reflects that the positioning method can adapt to the subjectivity and artificiality of vague location description.

In the case study, a group of users' viewpoints from Q&A on spatial cognition are transformed into the spatial scope in the real world. These spatial scopes can establish the relationship between qualitative spatial concepts and quantitative spatial data, so as to realize the representation of vague location description in GIS. Therefore, the query system with a Q&A mechanism embodies the application scenario of this method as a natural language interface. The case study results show that the supervaluation semantics can be adjusted in time according to the practical application, and the positioning results are precise and fit intuitive cognition. In the future, knowledge graphs will be introduced to further improve the semantic reasoning ability and positioning accuracy for vague location description.

**Author Contributions:** Conceptualization, Xueying Zhang and Peng Ye; data curation, Yulong Dang; formal analysis, Chunju Zhang; investigation, Peng Ye and Xueying Zhang; methodology, Peng Ye; project administration, Xueying Zhang; validation, Chunju Zhang; visualization, Peng Ye; writing—original draft, Peng Ye; writing—review and editing, Peng Ye and Yulong Dang All authors have read and agreed to the published version of the manuscript.

**Funding:** This research was funded by the National Natural Science Foundation of China (grant nos. 41631177, and 41971337), the Open Fund of Key Laboratory of Urban Land Resources Monitoring and Simulation, Ministry of Natural Resources (grant no. KF-2020-05-084), the Humanities and Social Sciences Foundation of Yangzhou University (grant no. xjj2021-08) and the Open Fund of Research Institute of Central Jiangsu Development, Yangzhou University (grant no. szfz202114).

**Institutional Review Board Statement:** Not applicable.

**Informed Consent Statement:** Informed consent was obtained from all subjects involved in the study.

**Data Availability Statement:** Not applicable.

**Acknowledgments:** The authors thank Mi Du and Xiao Zhang for their critical reviews and constructive comments.

**Conflicts of Interest:** The authors declare no conflict of interest.

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
