# Peer review of "Positioning Localities for Vague Spatial Location Description: A Supervaluation Semantics Approach"

_ijgi, doi:10.3390/ijgi11010068_

Round 1

Reviewer 1 Report

recent references could be included, the authors have only one reference in 2020 and two in 2021

more details could enhance the experiment section

Author Response

Point 1. Recent references could be included, the authors have only one reference in 2020 and two in 2021

Response 1: We thank the reviewer for the valuable comment. In the section of related work and discussion, the comparison and analysis of the method in this paper with the existing research have been added. Correspondingly, recent references have also been added, including the following:

  1. Xu, J.; Pan, X.; Zhao, J.; Fu, H. Virtual Reality-Based Fuzzy Spatial Relation Knowledge Extraction Method for Observ-er-Centered Vague Location Descriptions. ISPRS Int. J. Geo-Inf. 2021, 10, 833.
  2. Blaschke, T.; Merschdorf, H.; Cabrera-Barona, P.; Gao, S.; Papadakis, E.; Kovacs-Györi, A. Place versus Space: From Points, Lines and Polygons in GIS to Place-Based Representations Reflecting Language and Culture. ISPRS Int. J. Geo-Inf. 2018, 7, 452.
  3. Chen, S.; Zhang, H.; Yang, H. Urban Functional Zone Recognition Integrating Multisource Geographic Data. Remote Sens. 2021, 13, 4732.
  4. Liu, K.; Yin, L.; Lu, F.; Mou, N. Visualizing and exploring POI configurations of urban regions on POI-type semantic space. Cities 2020, 99, 102610.
  5. Xu, J.; Pan, X. A Fuzzy Spatial Region Extraction Model for Object’s Vague Location Description from Observer Perspective. ISPRS Int. J. Geo-Inf. 2020, 9, 703.

Point 2. More details could enhance the experiment section.

Response 2: The case study is the application of the methodology in an actual scenario. In order to make the case study clearer, on the one hand, the specific methodological description was supplemented; On the other hand, the citations of the methodological steps were added to the case study.

We added this point in revised manuscript and the detailed revision can be found in Line 497-507 Page 12, Line 522-532 Page 13, Line 546-551 Page 14, Line 635-637 Page 17 and Line 675-677 Page 18.

Reviewer 2 Report

Authors have attempted to make a position prescription method using supervaluation semantics. The paper looks interesting. 

1. In line 75-76, authors have stated that "Compared with other solutions....". But they have not provided the list of other solutions being used by researchers for addressing vaguness. Authors need to provide literature review for this. Authors have only mentioned two approaches (a) two-valued logic and (b) fuzzy logic. What about other methods? How many other methods are available at present? What is their application domain, limitations and challenges? these things need to be addressed by the authors for the benefit of readers. Authors need to provide little explanation about "Cone Model" and other models mentioned in line 105-107.

2.  Line 110-111 is not clear. Author need to provide more explanation for the benefit of reader.

3. In section 4.2.2, out of different thresholds of vague spatial objects, vague direction and vague distance, the MODIFIER is not clear. This need more explanation.

4. Authors may cite works on location vagueness studied by other authors, and may reveal what are the advantages and weakness of their proposed approach in relation to other studies.
https://www.mdpi.com/2220-9964/9/12/703/htm
https://agile-online.org/conference_paper/cds/agile_2013/posters/p_edoh-alove.pdf
https://link.springer.com/chapter/10.1007%2F3-540-26772-7_23
https://ieeexplore.ieee.org/document/4295469
https://ieeexplore.ieee.org/document/5277056

5. In the Discussion section authors need to be compare and link other studies. In this section, authors need to provide an unbiased evaluation of their findings and approaches with other work.

Author Response

Authors have attempted to make a position prescription method using supervaluation semantics. The paper looks interesting.

Point 1. In line 75-76, authors have stated that "Compared with other solutions....". But they have not provided the list of other solutions being used by researchers for addressing vaguness. Authors need to provide literature review for this. Authors have only mentioned two approaches (a) two-valued logic and (b) fuzzy logic. What about other methods? How many other methods are available at present? What is their application domain, limitations and challenges? these things need to be addressed by the authors for the benefit of readers. Authors need to provide little explanation about "Cone Model" and other models mentioned in line 105-107.

Response 1: We thank the reviewer for the valuable comment. In the related work section, we added a review and summary of the methods related to the topic of this paper. On the one hand, the methods mentioned in this paper, supplement each method of explanations, advantages and disadvantages. On the other hand, the other positioning methods have been complemented. In addition to the two main positioning methods (two-valued logic and fuzzy logic), the point-radius method, the egg-folk model and the spatial clustering method are also proposed. The detailed revision can be found in Line 165-205, Page 4.

For the explanation about "Cone Model" and other models that need to be added, the principles and related explanations of these methods have been supplemented in Table 1, Page 5.

Point 2. Line 110-111 is not clear. Author need to provide more explanation for the benefit of reader.

Response2: Both the 4-intersection model and the 9-intersection model are classic methods for modeling topological relationships. However, both models have disadvantages. For the 4-intersection model, there are many cases that are clearly distinguishable by people, but this model is powerless. For the 9-intersection model, there is not much improvement in the representation of the spatial relationship of plane-plane, point-point, point-line, and point-plane.

On this basis, the 9-intersection model (9IM) has been extended to 2 new models: 9-intersection model based on dimension extension and 9-intersection model based on Voronoi graph. For the 9-intersection model based on dimension extension, use the dimension expansion method to expand 9IM. The dimensionality of the intersection between the boundary, interior and complement of the point, line, and plane is used as the framework for the description of the spatial relationship. For the 9-intersection model based on Voronoi graph, the Voronoi region is used to replace the "complement" of the spatial target in the 9IM, and a nine tuples model of spatial relationships based on Voronoi is developed.

We added this point in revised manuscript and the detailed revision can be found in Table 1, Page 5.

Point 3. In section 4.2.2, out of different thresholds of vague spatial objects, vague direction and vague distance, the MODIFIER is not clear. This need more explanation.

Response 3: We thank the reviewer for the valuable comment. Modifiers in the context are important factors that affect the threshold of observation. The thresholds of vague spatial object, direction relation and distance relation are all affected by modifiers. For instance, for "near", "peripheral", "opposite" and other similar modifiers, will expand the threshold set in varying degrees of the buffer zone.

In addition, the effect of modifiers on the threshold needs to be determined. The specific threshold value, which is affected by the modifier, can be determined by the question and answer method. Explain the vagueness of modifier words based on quantifiable answers, establish the mapping relationship model between modifier words and vague, and set the threshold range on this basis.

We added this point in revised manuscript and the detailed revision can be found in Line 497-507 Page 13, Line 522-532 Page 14 and Line 546-551 Page 15.

Point 4. Authors may cite works on location vagueness studied by other authors, and may reveal what are the advantages and weakness of their proposed approach in relation to other studies.

https://www.mdpi.com/2220-9964/9/12/703/htm

https://agile-online.org/conference_paper/cds/agile_2013/posters/p_edoh-alove.pdf

https://link.springer.com/chapter/10.1007%2F3-540-26772-7_23

https://ieeexplore.ieee.org/document/4295469

https://ieeexplore.ieee.org/document/5277056

Response 4: We agree with the reviewer's comment. These relevant studies have been cited in the paper. The detailed revision can be found in Line 728-747 Page 22, Reference 66-70 Page 26.

  1. Straccia, U. Towards spatial reasoning in fuzzy description logics. In Proceedings of 2009 IEEE International Conference on Fuzzy Systems, 2009, pp. 512-517, doi: 10.1109/FUZZY.2009.5277056.
  2. Edoh-Alove, E.; Bimonte, S.; Pinet, F., Bédard, Y. Exploiting Spatial Vagueness in Spatial OLAP: Towards a New Hybrid Risk-Aware Design Approach. In Proceedings of 16th AGILE International Conference on Geographic Information Science, Leuven, Belgium, May 2013.
  3. Dilo, A.; de By, R.A.; Stein, A. Metrics for vague spatial objects based on the concept of mass. In Proceedings of 2007 IEEE International Fuzzy Systems Conference, 2007, pp. 1-6, doi: 10.1109/FUZZY.2007.4295469.
  4. Cheng, T.; Fisher, P.; Li, Z. Double Vagueness: Effect of Scale on the Modelling of Fuzzy Spatial Objects. In Proceedings of Developments in Spatial Data Handling. Springer, Berlin, Heidelberg, 2005.
  5. Xu, J.; Pan, X. A Fuzzy Spatial Region Extraction Model for Object’s Vague Location Description from Observer Perspective. ISPRS Int. J. Geo-Inf. 2020, 9, 703.

Point 5. In the Discussion section authors need to be compare and link other studies. In this section, authors need to provide an unbiased evaluation of their findings and approaches with other work.

Response 5: We thank the reviewer for the valuable comment. At present, many studies attempt to investigate for the public, multi-scale, large-scale, in order to obtain the universal spatial cognitive law, and to provide a realistic basis for establishing the mainstream mapping relationship between vague description and real space.

In fact, everyone’s perception of the spatial location is subjective, and the expression of location description is artificial. The same location description refers to different real spaces due to different scenes and individuals. The method proposed in this paper can describe the vagueness in the location description precisely and intuitively.

We added this point in revised manuscript and the detailed revision can be found in Line 728-757, Page 22.

Reviewer 3 Report

The paper produces a novel idea and has a clear presentation of contents. Usage of the super valuation semantics to detect the location adds the required novelty. . Knowledge graph transformations with the semantic parsing and  positioning of the location description aimed with accurate precisions  integrated into intelligence, forms the crux of this article.

Author Response

The paper produces a novel idea and has a clear presentation of contents. Usage of the super valuation semantics to detect the location adds the required novelty. Knowledge graph transformations with the semantic parsing and positioning of the location description aimed with accurate precisions integrated into intelligence, forms the crux of this article.

Response: We agree with the reviewer's comment. In the future, the method proposed in this paper also needs to be extended to complex location description. Whereas, The complexity of location description increases its vagueness, and the factors considered in the process of precisification based on supervaluation semantics are more di-versified. Therefore, the rules of the confirmation of the conditional cut-off points and the multi-conditional compound computation are more complex, which need the specific prior and domain knowledge to guide the precisification. At present, knowledge graph leads knowledge engineering of big data into a brand-new stage, which has advantages in computational and inferential aspects, and becomes an important tool of knowledge management and service. Knowledge graph transforms the semantic parsing and precise positioning of the location description into intelligence, which is also facing challenges in the field of human-computer cooperation and complex reasoning.

Reviewer 4 Report

This paper is to explore positioning localities for vague spatial location  description using a supervaluation semantics approach. The topic is interesting.  Overall study processes are sound and valid. Study results are proper. Followings are some comments to improve the quality of the paper.

  • Address a study necessity clearly in the introduction section.
  • Address a study purpose clearly in the introduction section
  • Enhance figure 1. You may delete a children figures. It does not look like a professional. 
  • In figure 2, vague predicate => vague predication 
  •  Add a time period to make this case study in the case study section. 
  • Double check some typos. 

Author Response

This paper is to explore positioning localities for vague spatial location description using a supervaluation semantics approach. The topic is interesting. Overall study processes are sound and valid. Study results are proper. Followings are some comments to improve the quality of the paper.

Point 1. Address a study necessity clearly in the introduction section.

Response 1: We thank the reviewer for the valuable comment. The quantitative direction, coordinate and region can be represented directly by points, lines and polygons in the geographic information system (GIS). Whereas, it is difficult for GIS to directly deal with the qualitative location description with vague-ness. In fact, location descriptions are widely found in witness records, social media, historical documents and so on. Therefore, it is necessary to analyze the spatial information in vague location description so that it can be represented in GIS.

We added this point in revised manuscript and the detailed revision can be found in Line 84-90, Page 2.

Point 2. Address a study purpose clearly in the introduction section.

Response2: We thank the reviewer for the valuable comment. This paper presents a positioning method based on supervaluation semantics. This method can transform the location description containing a set of spatial cognition views into real-world spatial scope, and realize the representation of vague location description in GIS.

We added this point in revised manuscript and the detailed revision can be found in Line 90-92, Page 2.

Point 3. Enhance figure 1. You may delete a children figures. It does not look like a professional.

Response 3: Figure 1 has been modified. Replaced the children’s picture in the figure. The detailed revision can be found in Figure 1, Page 8.

Point 4. In figure 2, vague predicate => vague predication

Response 4: We agree with the reviewer's comment. The “vague predicate” has been replaced with “vague predication”. The detailed revision can be found in Figure 2, Page 10.

Point 5. Add a time period to make this case study in the case study section.

Response 5: In order to analyze the application effect of the method proposed in this paper, the Nanjing Road Walkway in Shanghai City in 2021 is selected as a case to illustrate the experiment. We added this point in revised manuscript and the detailed revision can be found in Line 610-612, Page 16.

Point 6. Double check some typos.

Response 6: We agree with the reviewer's comment. The language expression of the whole paper has been checked and revised.

Reviewer 5 Report

My comments:

  1. abstract, please explain acronym Q&A? 
  2. abstract, please add information about obtained results from research test.
  3. main body of text, please check if all acronyms are explained.
  4. chapter 2, where are conclusions from scientific knowledge analysis?  And also where is a paragraph about novelty of paper? It must be added.
  5. equations (1-15), all symbols must be explained in text, please check it.
  6. discussion, please also compare your solution with scientific knowledge analysis. Why your solution is better? 
  7. conclusion, it must be revised. Please add more information about novelty of paper, about obtained results.

Author Response

My comments:

Point 1. Abstract, please explain acronym Q&A?

Response 1: We have replaced “Q&A” with “question answering system”. The detailed revision can be found in Line 25, Page 1.

Point 2. Abstract, please add information about obtained results from research test.

Response2: We thank the reviewer for the valuable comment. The case can verify the transformation of a set of users' viewpoints on spatial cognition into the real-world spatial scope, to realize the representation of vague location description in the geo-graphic information system. The result shows that the method proposed in the paper breaks through the traditional vagueness modeling which only focuses on spatial relationship, and enhances the interpretability of semantics of vague location description. Moreover, supervaluation semantics can obtain the precisification results of vague location description in different situations, and the positioning localities are more suitable to individual subjective cognition.

We added this point in revised manuscript and the detailed revision can be found in Line 26-32, Page 1.

Point 3. Main body of text, please check if all acronyms are explained.

Response 3: The language expression of the whole paper has been checked and revised. Added explanations to acronyms including GIS, POI, etc.

Point 4. Chapter 2, where are conclusions from scientific knowledge analysis?  And also where is a paragraph about novelty of paper? It must be added.

Response 4: We thank the reviewer for the valuable comment. The main innovation of this study is reflected in the following two aspects:

(1) Construction of vague location description representation model considering multi-factors. Based on the cognitive mechanism of spatial location, a unified framework of spatial location description is proposed, which defines three cognitive factors: Reference frame, spatial object and spatial relationship. The framework is used to illustrate the vagueness of different cognitive and abstract levels in the location description. Different from the traditional spatial information modeling which focuses on spatial relationship, this paper establishes the vagueness relation and influence among different information factors by the strategy of multi-factors representation. It not only combs the source of vagueness in location description more comprehensively, but also enhances the interpretability of semantics of vague location description.

(2) A supervaluationist theory-based positioning method for vague spatial location description is proposed. Based on the basic principles of supervaluation semantics, precisification models of vague predicates are constructed for spatial object and spatial relationship. From the three aspects of extension, anti-extension and penumbra, the threshold range of the cut-off point in the precisification model is set according to the location description context, so as to obtain precisification results in different situations. Location description as the expression of the result of personal spatial cognition is strongly subjective. In this method, different contexts are supervalued separately, making the inferred spatial location in real world more suitable for personalized subjective cognition.

We added this point in revised manuscript and the detailed revision can be found in Line 92-112, Page 2.

Point 5. Equations (1-15), all symbols must be explained in text, please check it.

Response 5: Check the whole of equations, and provide supplementary explanations for some of the symbols in the equations.

Point 6. Discussion, please also compare your solution with scientific knowledge analysis. Why your solution is better?

Response 6: We thank the reviewer for the valuable comment. At present, many studies attempt to investigate for the public, multi-scale, large-scale, in order to obtain the universal spatial cognitive law, and to provide a realistic basis for establishing the mainstream mapping relationship between vague description and real space.

In fact, everyone’s perception of the spatial location is subjective, and the expression of location description is artificial. The same location description refers to different real spaces due to different scenes and individuals. The method proposed in this paper can describe the vagueness in the location description precisely and intuitively.

We added this point in revised manuscript and the detailed revision can be found in Line 728-757, Page 22.

Point 7. Conclusion, it must be revised. Please add more information about novelty of paper, about obtained results.

Response 7: We thank the reviewer for the valuable comment. In this paper, a positioning method of vague location description based on supervaluation semantics is proposed. On the one hand, the vagueness of location description is analyzed from multi-factors perspective. The traditional vague modeling of spatial relationship is extended to the modeling that takes into account the three aspects of reference frame, spatial objects and spatial relationships, which enhances the semantic analysis ability for the vagueness of location description. On the other hand, based on the spatial positioning of vague location descriptions, the spatial information in the description is converted into the spatial range that can be represented in GIS. Based on the supervaluation theory, the precisification results of location description in different situations are obtained respectively, which reflects that the positioning method can adapt to the subjectivity and artificiality of vague location description.

In the case study, a group of users’ viewpoints from Q&A on spatial cognition are transformed into the spatial scope in the real world. These spatial scopes can establish the relationship between qualitative spatial concepts and quantitative spatial data, so as to realize the representation of vague location description in GIS. Therefore, the query system with Q&A mechanism embodies the application scenario of this method as a natural language interface. The case study results show that the supervaluation semantics can be adjusted in time according to the practical application, and the positioning results are precise and fit intuitive cognition.

We added this point in revised manuscript and the detailed revision can be found in Line 784-803, Page 23.

Round 2

Reviewer 5 Report

I accept the paper in current form.